# Endogenous RNAi pathway evolutionarily shapes the destiny of the antisense lncRNAs transcriptome

Ugo Szachnowski*, Sara Andjus* ⓘ, Dominika Foretek ⓘ, Antonin Morillon ⓘ, Maxime Wery ⓘ

**Antisense long noncoding (aslnc)RNAs are extensively degraded by the nuclear exosome and the cytoplasmic exoribonuclease Xrn1 in the budding yeast *Saccharomyces cerevisiae*, lacking RNAi. Whether the ribonuclease III Dicer affects aslncRNAs in close RNAi-capable relatives remains unknown. Using genome-wide RNA profiling, here we show that aslncRNAs are primarily targeted by the exosome and Xrn1 in the RNAi-capable budding yeast *Naumovozyma castellii*, Dicer only affecting Xrn1-sensitive aslncRNAs levels in Xrn1-deficient cells. The *dcr1* and *xrn1* mutants display synergic growth defects, indicating that Dicer becomes critical in the absence of Xrn1. Small RNA sequencing showed that Dicer processes aslncRNAs into small RNAs, with a preference for Xrn1-sensitive aslncRNAs. Consistently, Dicer localizes into the cytoplasm. Finally, we observed an expansion of the exosome-sensitive antisense transcriptome in *N. castellii* compared with *S. cerevisiae*, suggesting that the presence of cytoplasmic RNAi has reinforced the nuclear RNA surveillance machinery to temper aslncRNAs expression. Our data provide fundamental insights into aslncRNAs metabolism and open perspectives into the possible evolutionary contribution of RNAi in shaping the aslncRNAs transcriptome.**

## Introduction

Initially considered as by-products of the pervasive transcription of eukaryotic genomes, long noncoding (lnc)RNAs have been progressively recognized as genuine transcripts playing important roles in the regulation of multiple cellular processes (Mercer et al, 2009; Wery et al, 2011; Rinn & Chang, 2012; Jarroux et al, 2017). Supporting the idea that lncRNAs can be functionally important, the dysregulated expression of some of them has been associated to diseases, including cancer and neurological disorders (Schmitt & Chang, 2016; Renganathan & Felley-Bosco, 2017; Saha et al, 2017; Schmitt & Chang, 2017).

Different classes of lncRNAs have been described (Jarroux et al, 2017). Among them, the "antisense" (as)lncRNAs are synthesized from

the strand opposite to "sense" protein-coding genes (Pelechano & Steinmetz, 2013) and have attracted a lot of attention given their potential to regulate gene expression (Kopp & Mendell, 2018). In fact, examples of aslncRNA-mediated regulation of gene expression have been reported in different organisms, including the budding yeast *Saccharomyces cerevisiae* (Camblong et al, 2007, 2009; Uhler et al, 2007; Berretta et al, 2008; Houseley et al, 2008; Pinskaya et al, 2009; Van Dijk et al, 2011; van Werven et al, 2012), the fission yeast *Schizosaccharomyces pombe* (Wery et al, 2018a), plants (Swiezewski et al, 2009), and Mammals (Lee & Lu, 1999; Yap et al, 2010).

One of the most striking features of aslncRNAs is their low cellular abundance. Pioneer works in *S. cerevisiae* have revealed that they are extensively degraded by RNA surveillance machineries (Tisseur et al, 2011; Tudek et al, 2015). Consequently, these "cryptic" aslncRNAs cannot be detected in wild-type (WT) cells but accumulate upon inactivation of the factor responsible for their degradation. For example, the cryptic unstable transcripts (CUTs) accumulate in cells lacking Rrp6 (Wyers et al, 2005; Neil et al, 2009; Xu et al, 2009), a nonessential 3′-5′ exoribonuclease of the nuclear exosome (Houseley et al, 2006). On the other hand, the Xrn1-sensitive unstable transcripts (XUTs) are degraded by the cytoplasmic 5′-3′ exoribonuclease Xrn1 (Van Dijk et al, 2011). Despite some of them are produced from intergenic regions, most CUTs and XUTs are antisense to protein-coding genes, at least partially.

This classification into CUTs or XUTs is informative as it provides insights into the RNA decay pathway by which they are degraded. However, it is not exclusive, and there is a non-negligible overlap between the two classes (Van Dijk et al, 2011; Wery et al, 2016). Indeed, the nuclear and the cytoplasmic RNA surveillance pathways can cooperate to target the same transcript, so that a CUT that would escape the nuclear degradation can be targeted by Xrn1 once exported in the cytoplasm. Alternatively, but not exclusively, overlapping lncRNA isoforms produced from the same transcription unit can be degraded by different RNA surveillance pathways (Marquardt et al, 2011).

Both Rrp6 and Xrn1 are conserved across eukaryotes (Houseley et al, 2006; Nagarajan et al, 2013). In this respect, CUTs and XUTs were recently identified in fission yeast (Atkinson et al, 2018; Watts et al, 2018; Wery et al, 2018b), and they are also mainly antisense to

---

ncRNA, Epigenetic and Genome Fluidity, Institut Curie, Sorbonne Université, CNRS UMR 3244, Paris, France

Correspondence: antonin.morillon@curie.fr; maxime.wery@curie.fr
*Ugo Szachnowski and Sara Andjus contributed equally to this work.

 https://doi.org/10.26508/lsa.201900407 vol 2 | no 5 | e201900407 **1 of 12**

protein-coding genes in this species. This indicates that the roles of the nuclear exosome and Xrn1 in restricting aslncRNAs levels have been conserved across the yeast clade.

However, one singularity that distinguishes *S. cerevisiae* from most other eukaryotes is the loss of the RNAi system during evolution, so it lacks the ribonuclease III Dicer that can process double-stranded (ds)RNA structures into siRNAs (Drinnenberg et al, 2009). However, upon heterologous expression in *S. cerevisiae* of RNAi factors from the close RNAi-capable relative species *Naumovozyma castellii* (Drinnenberg et al, 2009, 2011), we observed a massive production of siRNAs from asXUTs, indicating that they can form dsRNA structures with their paired-sense mRNAs in vivo (Wery et al, 2016). Consistent with this observation, *N. castellii* Dicer was detected in the cytoplasm when expressed in *S. cerevisiae* (Cruz & Houseley, 2014).

*S. pombe* has a functional RNAi machinery (Volpe et al, 2002), but asXUTs are insulated from it (Wery et al, 2018b). This is probably explained by different subcellular localization, as Dicer is restricted to the nucleus in fission yeast, mainly contributing in heterochromatin formation at centromeric repeats (Woolcock & Buhler, 2013). Yet, Dicer was shown to control a novel class of lncRNAs, referred to as Dicer-sensitive unstable transcripts (DUTs), which are also mainly antisense to protein-coding genes (Atkinson et al, 2018). Thus, in fission yeast, Dicer contributes to the control of aslncRNAs levels.

The discovery of RNAi in budding yeasts, such as *N. castellii*, *Kluyveromyces polysporus*, and *Candida albicans*, is quite recent (Drinnenberg et al, 2009) and whether RNAi plays any role in aslncRNAs metabolism in these species remains largely unknown. In this context, it has recently been proposed that the loss of RNAi in *S. cerevisiae* could have led to an expansion of the aslncRNAs transcriptome (Alcid & Tsukiyama, 2016). This hypothesis was essentially based on the observation that aslncRNAs expression levels, length, and degree of overlap with the paired-sense protein-coding genes are globally reduced in *N. castellii* compared with *S. cerevisiae*. However, these analyses were performed in WT strains, in which most aslncRNAs are likely to be degraded. Furthermore, it was not experimentally demonstrated that aslncRNAs are targeted by the endogenous RNAi machinery in *N. castellii*.

Here, we addressed the question of aslncRNAs degradation in *N. castellii* (Drinnenberg et al, 2009). Using deep transcriptome profiling in mutants of *DCR1*, *XRN1*, and *RRP6*, we showed that aslncRNAs are primarily degraded by the exosome and Xrn1. The loss of Dicer leads to a weak but significant increase in global aslncRNAs levels when combined to the *xrn1* mutation, suggesting that Dicer might become critical in the absence of Xrn1. This idea is supported by genetic evidence showing that the *dcr1* and *xrn1* mutants display synergic growth defects. Using small RNA sequencing, we showed that Dicer can process aslncRNAs into small RNAs, with a preference for asXUTs. Consistently, immunofluorescence experiments revealed that Dicer localizes in the cytoplasm. Finally, comparative analyses between aslncRNAs from *N. castellii* and *S. cerevisiae* revealed an expansion of the exosome-sensitive antisense transcriptome in the RNAi-capable budding yeast, suggesting that the nuclear RNA surveillance machinery has been evolutionarily reinforced for the control of aslncRNAs expression in a context where a Dicer-dependent ribonuclease III activity is present in the cytoplasm, possibly to prevent uncontrolled siRNAs

production. Together, our data provide fundamental insights into the aslncRNAs metabolism in a yeast species endowed with cytoplasmic RNAi, further highlighting the conserved roles of the exosome and Xrn1 in the control of aslncRNAs levels in eukaryotes.

# Results

## AslncRNAs are primarily degraded by Rrp6 and Xrn1 in *N. castellii*

To characterize the population of aslncRNAs in *N. castellii*, we performed genome-wide RNA profiling using RNA-seq data obtained from WT, *dcr1Δ*, *xrn1Δ*, and *rrp6Δ* cells (Fig 1A). For the identification of DUTs and XUTs, we performed RNA-Seq in WT, *dcr1Δ*, and *xrn1Δ* strains, followed by segmentation using the algorithm that we previously developed to annotate CUTs (Watts et al, 2018) and XUTs (Wery et al, 2016, 2018b) in other yeast species. For the identification of *N. castellii* CUTs, we profiled in parallel published RNA-Seq data obtained from *rrp6Δ* cells (Alcid & Tsukiyama, 2016). Among all the ≥200-nt segments not overlapping a coding sequence, tRNA, sn(o)RNA or rRNA on the same strand, using a signal threshold and differential expression analysis between each mutant and its corresponding WT control (Fig 1A; see the Materials and Methods section), we identified 146 stable unannotated transcripts (SUTs, i.e., lncRNAs detected in the WT context but not significantly stabilized in any of the mutant), 10 DUTs, 1,021 XUTs, and 1,280 CUTs (Figs 1B and S1A–C).

At the first glance, the number of DUTs appears to be dramatically low compared with CUTs and XUTs, indicating than the effect of Dcr1 on the lncRNAs transcriptome of *N. castellii* is marginal compared with Rrp6 and Xrn1 (Fig 1C–E). Moreover, these DUTs were also all identified as XUTs, and they are even more sensitive to Xrn1 than to Dcr1 (Fig S1D). Consequently, these 10 lncRNAs, sensitive to both Dcr1 and Xrn1, will only be considered as XUTs hereafter.

As previously observed in *S. cerevisiae* and *S. pombe* (Wery et al 2016, 2018b; Atkinson et al, 2018), many lncRNAs are targeted by both Rrp6 and Xrn1 in *N. castellii* (Fig 1B). Consistently, CUTs and XUTs globally display a moderate sensitivity to Xrn1 and Rrp6, respectively (Fig 1D and E). More precisely, 426 XUTs are stabilized upon inactivation of Rrp6 (Table S1; *rrp6Δ*/WT ratio >2, *P* < 0.05), whereas 610 CUTs accumulate in the absence of Xrn1 (Table S2; *xrn1Δ*/WT ratio >2, *P* < 0.05). Furthermore, 232 CUTs overlap ≥50% of a XUT (Fig S1E). This indicates that Rrp6 and Xrn1 also cooperate to restrict lncRNAs levels in *N. castellii*.

Most of the transcripts we identified are novel (Fig S1F) and are antisense to protein-coding genes, including 93 SUTs (64%), 622 XUTs (61%), and 868 CUTs (68%). These proportions increase when taking into account all the transcripts annotated in *N. castellii* and not only the coding sequences (Fig S1G). Interestingly, we observed that the solo lncRNAs (i.e., those that are not antisense) are globally more expressed than the antisense ones. This is not only the case for the SUT, XUT, and CUT classes in WT cells (Fig 1F) but also for XUTs and CUTs in *xrn1Δ* and *rrp6Δ* cells, respectively (Fig S1H and I).

Overall, we annotated 2,247 lncRNAs in *N. castellii*, 1,583 of which are antisense to protein-coding genes. The vast majority of them are unstable and are primarily degraded by the nuclear exosome

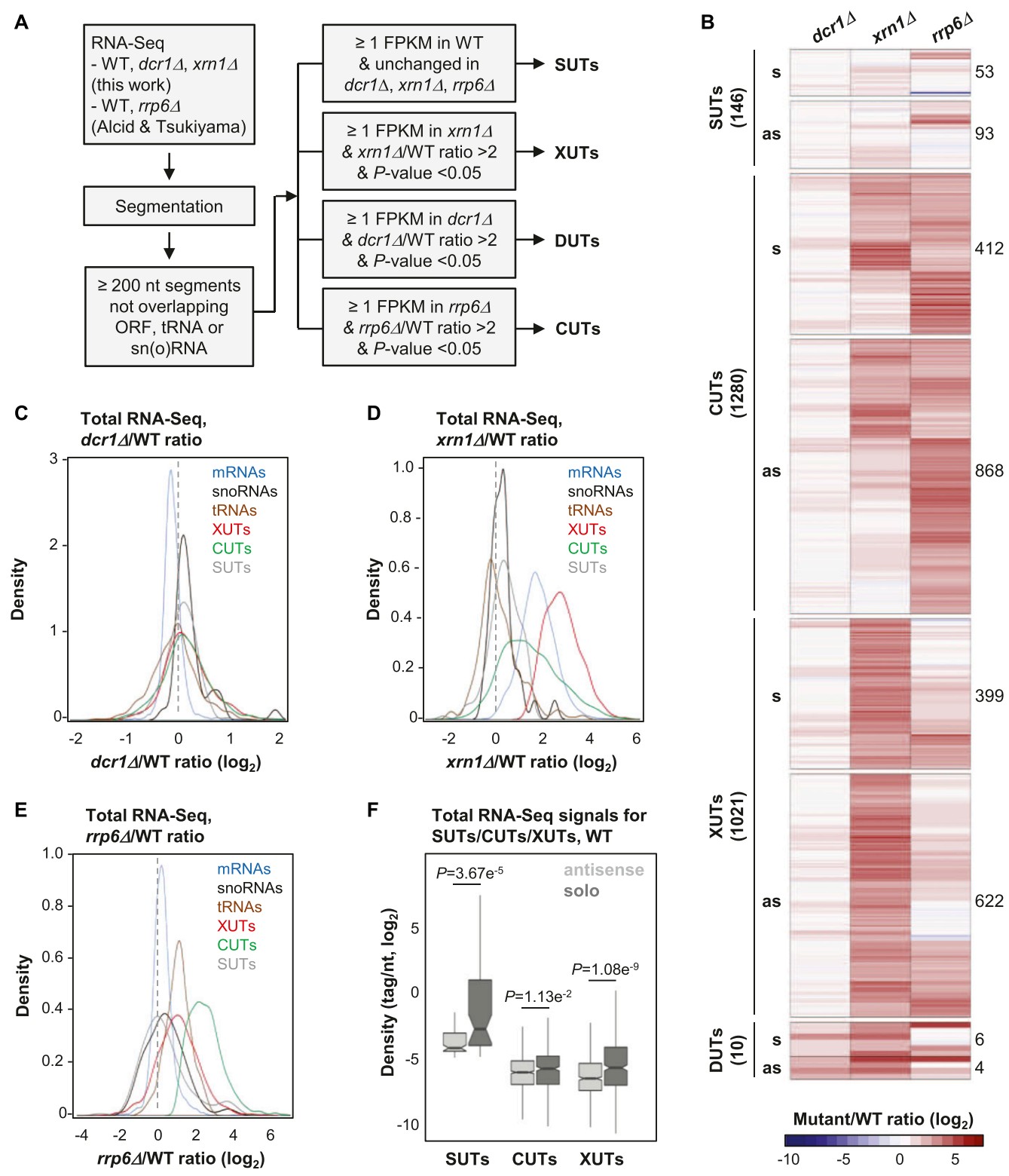

**Figure 1. AslncRNAs are primarily degraded by Xrn1 and Rrp6 in *N. castellii*.**
**(A)** Experimental strategy to annotate aslncRNAs in *N. castellii*. RNA-Seq data from biological duplicates of WT, *dcr1Δ*, and *xrn1Δ* cells were segmented using the ZINAR algorithm (Wery et al, 2016). Previously published RNA-Seq data from biological duplicates of *rrp6Δ* cells (Alcid & Tsukiyama, 2016) were segmented in parallel using the same tool. Among the ≥200-nt segments not overlapping an open reading frame (ORF), tRNA, or sn(o)RNAs, we identified 146 SUTs (signal in WT ≥ 1 FPKM; insensitive to Dcr1, Xrn1, or Rrp6), 1021 XUTs (signal in *xrn1Δ* ≥ 1 FPKM; *xrn1Δ*/WT ratio >2, *P* < 0.05), 10 DUTs (signal in *dcr1Δ* ≥ 1 FPKM; *dcr1Δ*/WT ratio >2, *P* < 0.05), and 1,280 CUTs (signal in *rrp6Δ* ≥ 1 FPKM; *rrp6Δ*/WT ratio >2, *P* < 0.05). **(B)** Heat map of the expression fold-change (ratio of tag densities, log$_2$ scale) for SUTs (146), CUTs (1280), XUTs (1021), and DUTs (10) in

(CUTs) and/or Xrn1 (XUTs), with almost no effect of Dcr1. Fig S2A and B shows snapshots of RNA-Seq signals for illustrative examples of asXUTs and asCUTs.

### dcr1 and xrn1 mutants are synergic

The data above indicate that Dcr1 has no major impact on aslncRNAs levels when Xrn1 and Rrp6 are functional (see Fig 1C). But is it also the case in cells lacking Xrn1 or Rrp6?

Globally, the loss of Dcr1 in xrn1Δ cells results into a moderate but significant increase in asXUTs levels compared with the single xrn1Δ mutant (Fig 2A; $P = 1.77 \times 10^{-5}$, Wilcoxon rank-sum test; see examples in Fig S2A–D), with no effect on the solo XUTs (Fig 2A; $P = 0.0633$, Wilcoxon rank-sum test).

In contrast, deleting DCR1 in rrp6Δ cells has no significant effect on global CUTs levels, independently of their solo or antisense configuration (Fig 2B; $P = 0.513$ and $0.991$, respectively; Wilcoxon rank-sum test).

The marginal effect of Dcr1 inactivation on the coding and noncoding transcriptomes (Figs 1B and C and S2E) is consistent with the normal growth of the dcr1Δ mutant, which is undistinguishable from the WT strain (Figs 2C and S2F). Interestingly, the growth of the dcr1Δ xrn1Δ double mutant is more affected than the xrn1Δ single mutant in rich medium at the optimal temperature 25°C (Figs 2C and S2F). This effect is even stronger at higher (32°C) or lower (18°C) temperatures, or when cells are grown on synthetic medium (Fig 2C).

Thus, Dcr1 significantly impacts asXUTs levels in xrn1Δ cells, and the dcr1 and xrn1 mutants display synergic growth defects, indicating that Dcr1 becomes critical when Xrn1 is not functional, consistent with the idea that Dcr1 and Xrn1 share similar substrates.

### AsXUTs are preferred aslncRNAs targets of Dicer for small RNAs production

We asked whether aslncRNAs are processed into small RNAs by Dicer in N. castellii. We sequenced small RNAs from WT, xrn1Δ, dcr1Δ, and xrn1Δ dcr1Δ cells.

In the WT and xrn1Δ strains, but not in dcr1Δ and xrn1Δ dcr1Δ, we observed the accumulation of 22–23-nt small RNAs, with U as the preferred 5' nucleotide (Fig S3A), which corresponds to the previously described features of siRNAs in N. castellii (Drinnenberg et al, 2009). Subsequent bioinformatics analyses filtering 22–23-nt small RNAs revealed that all classes of aslncRNAs are globally targeted by Dcr1 for small RNA production. In fact, small RNA densities are higher for the antisense SUTs, CUTs, and XUTs than their solo counterparts, especially in the xrn1Δ context (Figs 3A and S3B–D). Notably, this is also the case in the WT strain, indicating that aslncRNAs can be processed by Dcr1 when Rrp6 and Xrn1 are functional. This suggests that in WT cells, a fraction of aslncRNAs escape the RNA surveillance machineries and interact with the

paired-sense mRNAs to form dsRNA that can be processed by Dcr1 into small RNAs. Furthermore, in this condition, the asXUTs appear to be the preferred targets of Dcr1 among the three classes of aslncRNAs (Fig 3A). As illustrative examples, snapshots for the XUT0527/C05780 and XUT0213/A12460 pairs show that 22–23-nt small RNAs are produced from the asXUT/mRNA overlapping region in the WT context, with an increase in small RNAs densities in xrn1Δ (Figs 3B and S3E). In contrast, for the CUT0672/C05770 and CUT0275/A12440 pairs, the levels of 22–23-nt small RNAs in WT cells remain low (Figs 3B and S3E).

Together, these data show that aslncRNAs in N. castelli are efficiently targeted by Dcr1 for the production of small RNAs, with a preference for asXUTs.

### Dcr1 localizes in the cytoplasm

The observation that aslncRNAs are processed into small RNAs in N. castellii indicates that they can form dsRNA structures with the paired-sense mRNAs, which co-localize with Dcr1 into the same subcellular compartment. Because asXUTs (i.e., the aslncRNAs that are degraded in the cytoplasm) constitute the preferred targets of Dcr1 for small RNAs production, we anticipated that Dcr1 localizes in the cytoplasm. Further supporting this hypothesis, Dcr1 was previously detected as cytoplasmic foci when artificially expressed as a fusion with the GFP in S. cerevisiae (Cruz & Houseley, 2014).

We constructed a Dcr1-GFP strain in N. castellii (Fig S4A). Upon direct visualization in living cells, Dcr1-GFP appeared as individual discrete foci, which are absent not only in the untagged control strain but also in cells expressing the GFP alone (Fig S4B). When detected using GFP nanobody by immunofluorescence in fixed cells, these foci were found in the cytoplasm (Fig 4). Importantly, small RNA sequencing showed that the Dcr1-GFP fusion remains functional for the production of 22–23-nt small RNAs (Fig S4C).

From these observations, we conclude that Dcr1 localizes in the cytoplasmic compartment in N. castellii.

### Expansion of the exosome-sensitive aslncRNAs transcriptome in N. castellii

It has been recently proposed that RNAi could have evolutionarily contributed to restrict the aslncRNAs transcriptome in N. castellii (Alcid & Tsukiyama, 2016). This hypothesis was based, for instance, on the observation that 170 aslncRNAs annotated in a WT strain of N. castellii are shorter and display a reduced overlap with the paired-sense mRNAs in comparison with the set of aslncRNAs in S. cerevisiae (Alcid & Tsukiyama, 2016). As we considerably extended the repertoire of aslncRNAs in N. castelli, most of them being unstable because of their extensive degradation by Rrp6 and Xrn1, we decided to repeat this comparative analysis using our catalog of asCUTs and asXUTs. Note that some CUTs in S. cerevisiae are smaller

the dcr1Δ, xrn1Δ, and rrp6Δ mutants, relative to the corresponding WT strain. For each class of lncRNA, the number of antisense and solo (i.e., not antisense) transcripts is indicated. **(C)** Density plot of dcr1Δ/WT signal ratio for mRNAs (blue), sn(o)RNAs (black), tRNAs (brown), XUTs (red), CUTs (green), and SUTs (grey). **(D)** Density plot of xrn1Δ/WT signal ratio for the same classes of transcripts as above. **(E)** Density plot of rrp6Δ/WT signal ratio for the same classes of transcripts as above. **(F)** Box plot of densities (tag/nt, $\log_2$ scale) for the antisense (light grey) and solo (dark grey) SUTs, CUTs, and XUTs in WT cells. The P-values (adjusted for multiple testing with the Benjamini–Hochberg procedure) obtained upon two-sided Wilcoxon rank-sum test are indicated. Outliers: not shown.

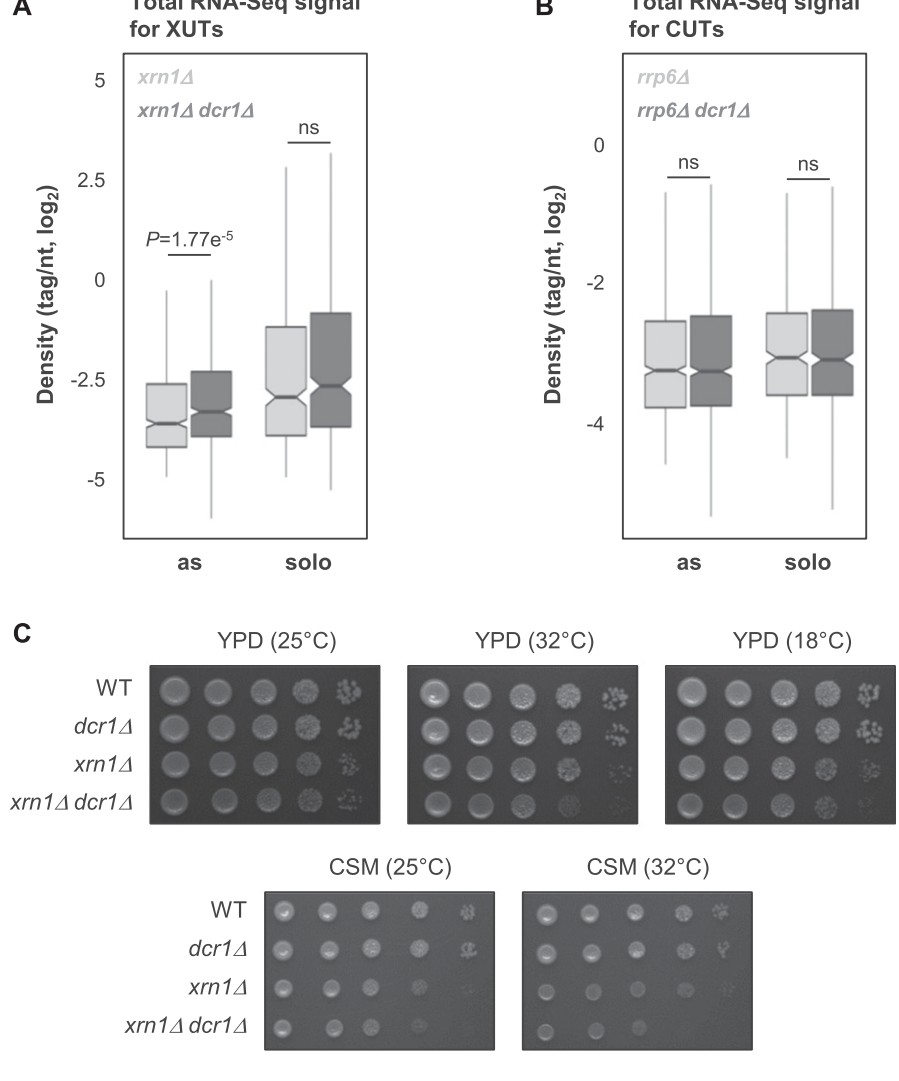

**Figure 2.  The *dcr1* and *xrn1* mutants display synergic defects.**
**(A)** Box plot of densities (tag/nt, log₂ scale) for the antisense (as) and solo XUTs in the *xrn1Δ* (light grey) and *xrn1Δ dcr1Δ* (dark grey) strains. The significant *P*-value (adjusted for multiple testing with the Benjamini–Hochberg procedure) obtained upon two-sided Wilcoxon rank-sum test is indicated. Outliers: not shown. Ns, not significant. **(B)** Box plot of densities (tag/nt, log₂ scale) for the antisense (as) and solo CUTs in the *rrp6Δ* (light grey) and *rrp6Δ dcr1Δ* (dark grey) strains. Data are presented as above. The raw RNA-Seq data have been previously published (Alcid & Tsukiyama, 2016). **(C)** Effects of *DCR1* and/or *XRN1* deletion on growth. Serial dilutions of YAM2478 (WT), YAM2795 (*dcr1Δ*), YAM2479 (*xrn1Δ*), and YAM2796 (*dcr1Δ xrn1Δ*) cells were dropped on rich medium (YPD) or CSM, then incubated at the indicated temperatures for 2–3 d.

than 200 nt (the threshold commonly used to define lncRNAs). We decided to remove all these <200-nt CUTs from our analysis, to avoid the introduction of a bias in the comparison based on the size of aslncRNAs.

We observed a weak but significant reduction of asCUTs size in *N. castellii* compared with *S. cerevisiae* (Fig 5A; median = 444 and 465 nt, respectively; *P* = 9.328 × 10⁻³, Wilcoxon rank-sum test). The size of asXUTs is also reduced in *N. castellii* (Fig 5B; median = 670 versus 709 nt in *S. cerevisiae*), but the difference is not significant (*P* = 0.3473, Wilcoxon rank-sum test). Surprisingly, we noted that the aslncRNAs annotated in this work are globally larger than the 170 previously annotated aslncRNAs (see Fig S5). As a possible explanation of this discrepancy, 54/170 (32%) of the previously annotated aslncRNAs are shorter than the commonly used 200-nt threshold (Fig S5).

Independently of the size of the aslncRNA, the degree of overlap with the paired-sense mRNA is probably more critical to determine its ability to form dsRNA. In this respect, we found no difference between the RNAi-capable and the RNAi-deficient species for the asCUTs (Fig 5C; median length of the overlap = 357 and 370 bp, respectively; *P* = 0.5044, Wilcoxon rank-sum test). In contrast, the overlap between asXUTs and their paired-sense genes is significantly reduced in *N. castellii* (Fig 5D; median = 400, versus 462 bp in *S. cerevisiae*; *P* = 1.315 × 10⁻⁴, Wilcoxon rank-sum test).

Finally, we analyzed the global coverage of the coding transcriptome by aslncRNAs (SUTs and/or CUTs and/or XUTs) in the two yeast species. Overall, aslncRNAs overlap 8.1% of the coding sequences in *N. castellii*, which is reduced in comparison with *S. cerevisiae* (12.9%). However, when we analyzed the asCUTs and asXUTs separately, we observed opposite patterns between the two species. Indeed, the coding transcriptome is mainly overlapped by asCUTs in the RNAi-capable species, whereas in *S. cerevisiae*, it is mainly covered by asXUTs (Fig 5E).

In conclusion, our analysis reveals an expansion of the exosome-sensitive aslncRNAs transcriptome in *N. castellii*, suggesting that the presence of Dicer in the cytoplasm has evolutionarily reinforced the nuclear RNA surveillance machinery to restrict the expression of aslncRNAs in the cytoplasmic compartment. Conversely, the loss of RNAi in *S. cerevisiae* might have allowed an expansion of the Xrn1-sensitive antisense transcriptome, relaxing the pressure to maintain aslncRNAs in the nucleus, away from Dcr1.

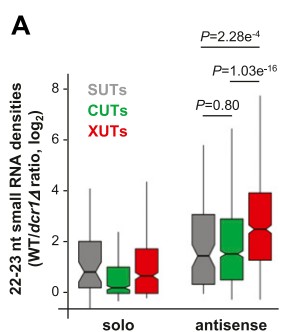
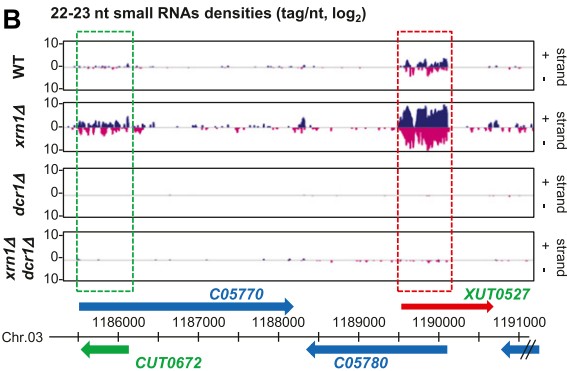

**Figure 3. AslncRNAs are processed into 22–23-nt small RNAs by Dcr1 in *N. castellii*.**

**(A)** Box plot of the WT/*dcr1Δ* ratio (log₂ scale) of 22–23-nt uniquely mapped small RNAs densities for the solo and antisense SUTs (grey), CUTs (green), and XUTs (red). The *P*-values (adjusted for multiple testing with the Benjamini–Hochberg procedure) obtained upon two-sided Wilcoxon rank-sum test are indicated. Outliers: not shown. **(B)** Snapshot of small RNAs densities for the *C05770/CUT0672* and *XUT0527/C05780* pairs. Densities of 22–23-nt small RNAs are shown in a separate panel for each strain. In each panel, signals (tag/nt, log₂) for the + and − strands are shown in blue and pink, respectively. The protein-coding genes, the CUT, and the XUT are represented by blue, green, and red arrows, respectively. The dashed boxes highlight the region of overlap between the aslncRNAs and the paired-sense mRNAs. The snapshot was produced using VING (Descrimes et al, 2015).

## Discussion

Previous works in the budding yeast *S. cerevisiae* and the fission yeast *S. pombe* have revealed that aslncRNAs are globally low abundant as they are extensively degraded by RNA surveillance machineries. For instance, the nuclear exosome targets a class of lncRNAs referred to as CUTs (Wyers et al, 2005; Neil et al, 2009; Xu et al, 2009), whereas the cytoplasmic 5′-3′ exoribonuclease Xrn1 degrades the so-called XUTs (Van Dijk et al, 2011), both types of transcripts being mainly antisense to protein-coding genes. However, this classification into CUTs and XUTs is not exclusive, some aslncRNAs being cooperatively targeted by the two RNA decay pathways. In fission yeast, an additional class of aslncRNAs (DUTs) was recently identified. DUTs accumulate in the absence of the ribonuclease III Dicer (Atkinson et al, 2018), highlighting the role of Dicer and RNAi in the control of aslncRNAs expression in fission yeast. This class of transcripts is absent in *S. cerevisiae*, which has lost the RNAi system during evolution. In this respect, *S. cerevisiae* is a notable exception among eukaryotes. In fact, a functional RNAi pathway was discovered in close relatives of *S. cerevisiae*, including *N. castellii* (Drinnenberg et al, 2009), a member of the *sensu lato* group of *Saccharomyces* that diverged from *S. cerevisiae* after the whole genome duplication (Cliften et al, 2006). The role of RNAi on aslncRNAs metabolism remains largely unknown in this species. However, a recent study proposed that the loss of RNAi in *S. cerevisiae* might have allowed the expansion of the aslncRNAs transcriptome (Alcid & Tsukiyama, 2016). This hypothesis was essentially based on the observation that aslncRNAs levels, length and degree of overlap with the paired-sense genes are reduced in the RNAi-capable budding yeast. However, these analyses were performed using a small set of aslncRNAs annotated from a WT strain of *N. castellii*, that is, a context in which most aslncRNAs are likely to be degraded. Furthermore, whether aslncRNAs are directly targeted by the RNAi machinery in *N. castellii* natural context remained unknown.

Using genome-wide RNA profiling in WT, *dcr1Δ*, *xrn1Δ*, and *rrp6Δ* strains of *N. castellii*, here we annotated 2,247 lncRNAs, including 1,583 aslncRNAs. Most of them are unstable and primarily degraded by the nuclear exosome (1,280 CUTs) and/or Xrn1 (1,021 XUTs), reinforcing the idea that the role of the 3′-5′ nuclear and 5′-3′ cytoplasmic RNA decay pathways in restricting aslncRNAs levels has

been conserved across the yeast clade. In contrast, the loss of Dcr1 has almost no effect on the aslncRNAs transcriptome. Only 10 DUTs accumulate in *dcr1Δ* cells, and they are also (even more) sensitive to Xrn1 (Fig S1D). This is marginal in comparison with the 1,392 DUTs annotated in fission yeast (Atkinson et al, 2018), raising the question of the function of Dcr1 in *N. castellii*.

*DCR1* has been conserved in some budding yeast species (Drinnenberg et al, 2009). However, deleting it in *N. castellii* confers no detectable growth defect, as shown under 50 different conditions (Drinnenberg et al, 2011). As previously proposed, the main role of Dcr1 in budding yeasts might be to silence retrotransposons (Drinnenberg et al, 2009). Consistently, although its genome still contains retrotransposons fragments, which constitute a major source for siRNAs production, no active retrotransposon has been identified in *N. castellii* (Drinnenberg et al, 2009). In addition, the expression of *N. castellii DCR1* and *AGO1* in *S. cerevisiae* leads to the silencing of endogenous retrotransposons (Drinnenberg et al, 2009), as well as to the loss of the dsRNA killer virus (Drinnenberg et al, 2011), with no other major impact on the transcriptome of *S. cerevisiae*.

However, several lines of evidence indicate that Dcr1 becomes critical in the absence of Xrn1. First, the global levels of asXUTs significantly increase in the *xrn1Δ dcr1Δ* mutant, compared with the *xrn1Δ* single mutant (Fig 2A). Second, the number of Dcr1-sensitive protein-coding genes is larger in the *xrn1Δ* context, in comparison with WT and *rrp6Δ* (Fig S2E). Third, the *dcr1Δ* and *xrn1Δ* mutants display synergic growth defects (Fig 2C). This indicates that the presence of Dcr1 becomes important for the cell viability in the absence of Xrn1, that is, when aslncRNAs accumulate in the cytoplasm, presumably forming dsRNA structures with the paired-sense mRNAs. In contrast, *DCR1* deletion was shown to suppress partially the growth defect of the *rrp6Δ* mutant (Alcid & Tsukiyama, 2016), indicating that Dcr1 is deleterious in Rrp6-lacking cells. Whether these opposite effects in the *xrn1Δ* and *rrp6Δ* backgrounds are related to siRNAs production from stabilized asXUTs and asCUTs, respectively, remains unknown. Additional analyses are required to decipher the molecular mechanisms underlying these genetic interactions.

The idea that Dcr1 and Xrn1 functionally interact is reinforced by the observation that Dcr1 localizes in the cytoplasm (Fig 4), which is consistent with previous observations made upon

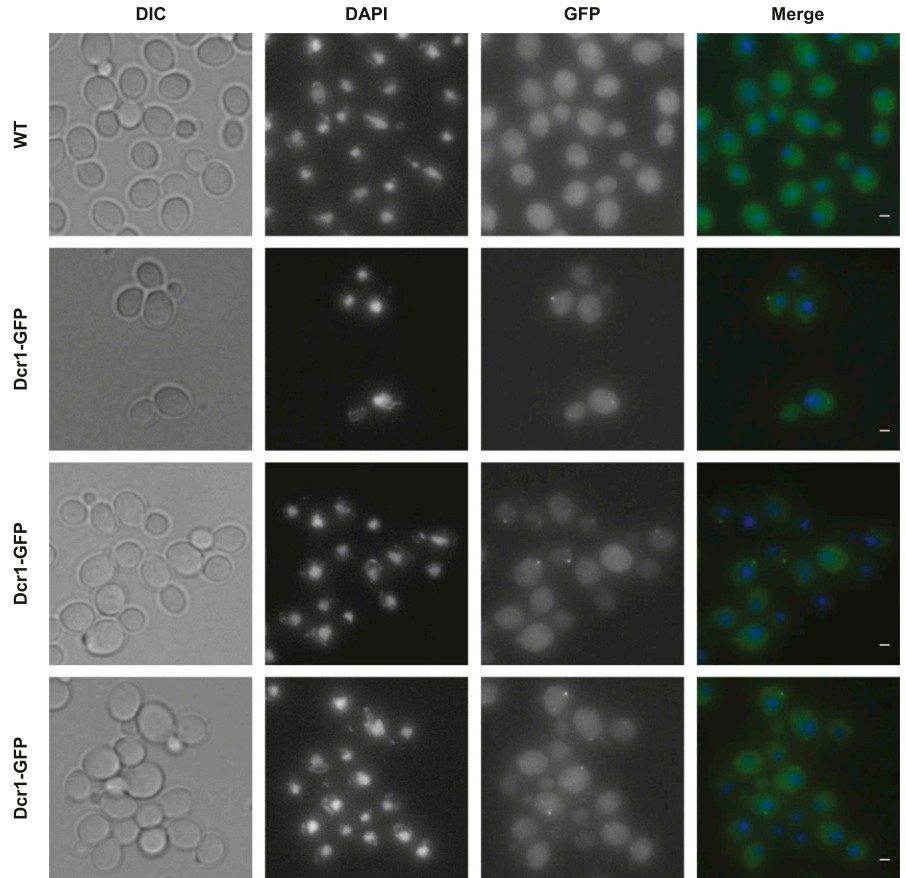

| DIC | DAPI | GFP | Merge |

**Figure 4. Subcellular localization of Dcr1 in *N. castellii*.**
YAM2478 (WT) and YAM2826 (Dcr1-GFP) cells were grown to mid-log phase in CSM, at 25°C. After fixation of cells, the subcellular localization of Dcr1-GFP was performed using immunofluorescence using GFP nanobody (see the Materials and Methods section). DAPI staining was used to visualize DNA. Scale bars: 1 μm.

expression of a Dcr1-GFP fusion in *S. cerevisiae* (Cruz & Houseley, 2014). Moreover, among the different classes of aslncRNAs, the asXUTs constitute the preferred target for small RNAs production (Fig 3A). Notably, these small RNAs are detected in WT cells, indicating that in this context, a fraction of asXUTs can escape Xrn1 to form dsRNA with the paired-sense mRNAs, which can then be processed by Dcr1 into small RNAs. To which extent the generated small RNAs are properly loaded into Argonaute to mediate post-transcriptional gene silencing remains unknown. The resulting effects, if any, are likely to be limited, in keeping with the absence of growth defects of the *dcr1Δ* mutant.

Besides asXUTs, asCUTs are also processed into small RNAs by Dcr1 (Fig 3A and B). As mentioned above, asCUTs (at least a fraction of them) could escape the degradation by Rrp6 and be exported to the cytoplasm. Then, as the asXUTs, they could be processed by Dcr1 upon dsRNA formation, if they are not degraded before by Xrn1. Alternatively, but not exclusively, we cannot exclude the possibility that a small amount of Dcr1 molecules in the cell localize in the nucleus, into levels that are under the detection threshold of our microscope. Perhaps, a more sensitive approach would help definitely answering the question of the subcellular localization of Dcr1 in RNAi-capable budding yeasts, even if all the current data are consistent with a cytoplasmic localization.

Recently, it has been proposed that the loss of RNAi in *S. cerevisiae* might have allowed the expansion of its aslncRNAs transcriptome (Alcid & Tsukiyama, 2016). Conversely, the conservation

of a functional RNAi machinery in *N. castellii* would have maintained a negative pressure against aslncRNAs. Among other observations, antisense expression at the *GAL10-GAL1* (*NCAS0E01670-NCAS0E01660*) locus was shown to be very low in WT cells of *N. castellii* (Alcid & Tsukiyama, 2016). Our RNA-Seq data confirmed this observation, further highlighting that despite the genomic organization of the *GAL1-GAL10-GAL7* locus has been conserved between *S. cerevisiae* and *N. castellii*, it is devoid of aslncRNA expression in RNAi-capable species, including in *xrn1Δ* and *rrp6Δ* strains (see the genome-browser associated to this work). Similarly, we confirm the absence of aslncRNA expression for the *PHO84* ortholog of *N. castellii* (*NCAS0B00220*). However, the differences between the RNAi-capable and RNAi-deficient species are more subtle than initially proposed. In fact, we show that more than 1,500 aslncRNAs co-exist with RNAi in *N. castellii*, mainly degraded by the exosome and Xrn1, representing an 8.1% cumulative overlap of the coding sequences by aslncRNAs, which is less than a twofold difference compared with *S. cerevisiae* (12.9%). Strikingly, when we analyzed the degree of overlap with the paired-sense ORFs, we observed that it is significantly reduced in *N. castellii* for the asXUTs but similar between the two species for the asCUTs (Fig 5C and D). Moreover, we observed that globally, the coding regions are mainly overlapped by asCUTs in the RNAi-capable species, whereas in *S. cerevisiae*, they are essentially overlapped by asXUTs. Together, our data suggest that the presence of an active RNAi machinery in the cytoplasm of *N. castellii* has favored the nuclear RNA decay pathway to restrict

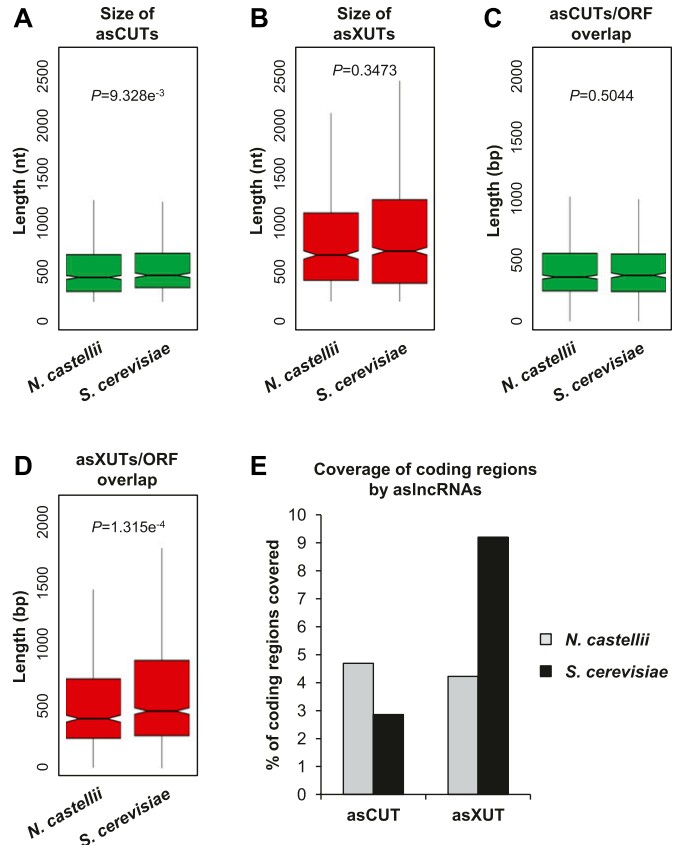

**Figure 5. Expansion of the exosome-sensitive aslncRNAs transcriptome in *N. castellii*.**
**(A)** Box plot of asCUTs size (nt) in *N. castellii* (n = 868) and *S. cerevisiae* (n = 535). For *S. cerevisiae*, all the <200-nt CUTs were removed from the analysis. The *P*-value obtained upon two-sided Wilcoxon rank-sum test is indicated. Outliers: not shown. **(B)** Same as above for asXUTs in *N. castellii* (n = 622) and *S. cerevisiae* (n = 1,152). **(C)** Box-plot of the overlap (bp) between asCUTs and the paired-sense ORF in *N. castellii* (n = 889) and *S. cerevisiae* (n = 574). For *S. cerevisiae*, all the <200-nt CUTs were removed from the analysis. The *P*-value obtained upon two-sided Wilcoxon rank-sum test is indicated. Outliers: not shown. **(D)** Same as above for asXUTs/ORF overlap in *N. castellii* (n = 674) and *S. cerevisiae* (n = 1,252). **(E)** Cumulative coverage of the coding regions by asCUTs and asXUTs in *N. castellii* (grey bars) and *S. cerevisiae* (black bars).

aslncRNAs expression, maybe to prevent uncontrolled and deleterious siRNAs production. This last hypothesis is supported by the observation that Dcr1 becomes deleterious in *rrp6Δ* cells (Alcid & Tsukiyama, 2016).

In conclusion, together with our previous studies in *S. cerevisiae* and *S. pombe*, this work in a budding yeast endowed with cytoplasmic RNAi provides fundamental insights into the metabolism and the decay of aslncRNAs in simple eukaryotic models. Our data not only further highlight the conserved roles of the nuclear exosome and Xrn1 in the control of aslncRNAs expression but also open perspectives into the possible evolutionary contribution of RNAi in shaping the aslncRNAs transcriptome. In this respect, the definition of the "cryptic" aslncRNAs landscape in organisms, such as plants and animals, where ribonuclease III activities are found in both the nucleus and the cytoplasm (Lee et al, 2003; Ha & Kim, 2014; Borges & Martienssen, 2015), will be of particular interest.

# Materials and Methods

## Strains, plasmids, and media

The genotypes of the strains used in this study are listed in Table S3. The YAM2478/DBP005 (WT) and YAM2795/DBP318 (*dcr1Δ*) strains were previously described (Drinnenberg et al, 2009).

The YAM2479 strain (*xrn1Δ::kanMX6*) was constructed by homologous recombination using the *kanMX6* marker flanked by long (>400 bp) *XRN1* targeting sequences. The *XRN1* ortholog in *N. castellii* is *C04170*, according to the Yeast Gene Order Browser (Byrne & Wolfe, 2005). The orthology was confirmed by CLUSTALO alignments (Fig S6A). To construct the *XRN1* deletion cassette, the *kanMX6* marker was first excised from the pFA6a-kanMX6 vector (Longtine et al, 1998) using BamHI and EcoRI digestion and cloned between the BamHI and EcoRI sites into the pCRII-TOPO plasmid (Invitrogen) to give the pCRII-kanMX6 plasmid. The 454-bp region upstream from *XRN1* was amplified by PCR using AMO1964-5 (see Table S4), and then cloned between the KpnI and BamHI sites into pCRII-kanMX6. Finally, the 481 bp downstream to *XRN1* were amplified by PCR using AMO1966-7 (see Table S4), and then cloned between the EcoRI and XbaI sites of the plasmid, giving the pAM376 vector. The deletion cassette was excised using KpnI–XbaI digestion and transformed into the YAM2478 strain. Transformants were selected on yeast extract–peptone–dextrose (YPD) + G418 plates at 25°C and screened by PCR on genomic DNA using oligonucleotides AMO1996-7. One clone was selected to give the YAM2479 strain, which was ultimately validated by Northern blot (Fig S6B).

To construct the YAM2796 strain (*dcr1Δ xrn1Δ*), the *xrn1Δ::kanMX6* cassette was amplified by PCR from YAM2479 genomic DNA using oligonucleotides AMO3227-8 (Table S4) and transformed into YAM2795. Transformants were selected and screened as above.

To construct the YAM2826 strain (Dcr1-GFP), the region corresponding to the last 478 bp of the *DCR1* ORF was amplified by PCR from YAM2478 genomic DNA using oligonucleotides AMO3323 and 3325 (Table S4). In parallel, the region corresponding to 525 bp after the stop codon of the *DCR1* ORF was amplified using oligonucleotides AMO3324 and 3326 (Table S4). After purification on agarose gel, the two PCR products displaying a 42-bp overlap were mixed and used as DNA templates for PCR using oligonucleotides AMO3323 and 3324. The resulting PCR product (1,047 bp long) was cloned between the KpnI and XbaI sites of the pCRII-TOPO plasmid (Invitrogen), to give the pCRII-Dcr1 vector. The GFP(S65T)-kanMX6 cassette was then amplified by PCR from the pFA6a-GFP(S65T)-kanMX6 plasmid using oligonucleotides AMO3327-8 (Table S4). The GFP(S65T)-kanMX6 PCR product was digested by BamHI and EcoRI and cloned between the same sites in the pCRII-Dcr1 vector, to give the pAM566 vector (pCRII-Dcr1-GFP-kanMX6). After verification of absence of mutation by Sanger sequencing, the Dcr1-GFP-kanMX6 construct was excised using NaeI digestion and transformed in the YAM2478 strain. Transformants were selected on YPD + G418 plates at 25°C and screened by PCR on genomic DNA using oligonucleotides AMO3229-30. One clone was selected and validated by Western blot (Fig S4A), giving the YAM2826 strain.

To construct the YAM2842 strain (*dcr1Δ::GFP-kanMX6*), the region corresponding to the *DCR1* promoter was amplified by PCR from

YAM2478 genomic DNA using oligonucleotides AMO3370-1 (Table S4). The resulting PCR product (470 bp) was purified and cloned between the KpnI and BamHI sites of the pAM566 plasmid, replacing the fragment corresponding to the end of the *DCR1* ORF, giving the pAM574 vector (pCRII-dcr1Δ::GFP-kanMX6). The absence of mutation was verified, then the *dcr1Δ::GFP-kanMX6* construct was excised and transformed in YAM2478 cells, as described above. Transformants were selected and screened as above. One clone was validated by Western blot (Fig S4A), giving the YAM2842 strain.

*N. castellii* strains were grown at 25°C in rich YPD medium to mid-log phase ($OD_{600}$ 0.5). For the microscopy analyses, the cells were grown under the same conditions in complete synthetic medium (CSM).

## Total RNA extraction

Total RNA was extracted from exponentially growing ($OD_{600}$ 0.5) cells using standard hot phenol procedure. RNA was resuspended in nuclease-free $H_2O$ (Ambion) and quantified using a NanoDrop 2000c spectrophotometer. Quality and integrity of extracted RNA was checked by Northern blot and/or analysis in a RNA 6000 Pico chip in a 2100 bioanalyzer (Agilent).

## Northern blot

10 μg of total RNA were loaded on denaturing 1.2% agarose gel and transferred to Hybond-XL nylon membrane (GE Healthcare). $^{32}$P-labelled oligonucleotides (see Table S4) were hybridized overnight at 42°C in ULTRAhyb-Oligo hybridization buffer (Ambion). For detection of the 5′ ITS1 fragment, a double-stranded DNA probe (obtained by PCR amplification using oligonucleotides AMO2002-2003) was $^{32}$P-labelled using the Prime-It II Random Primer Labeling Kit (Agilent), and then hybridized overnight at 65°C in PerfectHyb Plus Hybridization Buffer (Sigma-Aldrich).

## Total RNA-Seq

Total RNA-Seq analysis was performed from two biological replicates of YAM2478 (WT), YAM2479 (*xrn1Δ*), YAM2795 (*dcr1Δ*), and YAM2796 (*dcr1Δ xrn1Δ*) cells. For each sample, 1 μg of total RNA was mixed with 2 μl of 1:100 diluted ERCC RNA spike-in (Life Technologies), then ribosomal (r)RNAs were depleted using the RiboMinus Eukaryote v2 Kit (Life Technologies). Total RNA-Seq libraries were constructed from 50 ng of rRNA-depleted RNA using the TruSeq Stranded mRNA Sample Preparation Kit (Illumina). Paired-end sequencing (2 × 50 nt) was performed on a HiSeq 2500 system (Illumina).

The *N. castellii* reference genome was retrieved from version 7 of the Yeast Gene Order Browser (Byrne & Wolfe, 2005); snoRNAs were annotated using the *S. cerevisiae* snoRNAs as queries for blastn alignments (E value cutoff $e^{-8}$). Reads were mapped using version 2.0.9 of TopHat (Kim et al, 2013), with a tolerance of three mismatches and a maximum size for introns of 2 Kb. All bioinformatics analyses used uniquely mapped reads. Tag densities were normalized on the ERCC RNA spike-in signal.

## Annotation of lncRNAs

Segmentation was performed using the ZINAR algorithm (Wery et al, 2016). Briefly, the uniquely mapped reads from our WT, *dcr1Δ*, *xrn1Δ*, and *dcr1Δ xrn1Δ* samples were pooled. A signal was computed in a strand-specific manner for each position as the number of times it is covered by a read or the insert sequence between two paired reads. After $log_2$ transformation, the signal was smoothed using a sliding window (ranging from 5 to 200 nt, with 5-nt increment). All genomic regions showing a smoothed $log_2$ signal value above a threshold (ranging from 1.44 to 432, with 1.44 increments) were reported as segments. In total, 12,000 segmentations with different sliding window size and threshold parameters were tested in parallel, among which we arbitrarily selected one showing a good compromise between mRNA and novel lncRNAs detection. The parameters for the selected segmentation were: threshold = 27.36; sliding window size = 10 nt. Among the ≥200-nt novel segments that do not overlap ORF, tRNA or sn(o)RNA, we identified 1021 XUTs and 10 DUTs, showing a signal ≥1 FPKM (fragment per kilobase per million mapped reads) and >twofold enrichment in the *xrn1Δ* and *dcr1Δ* mutant, respectively, compared with the WT control, with a *P*-value < 0.05 (adjusted for multiple testing with the Benjamini–Hochberg procedure) upon differential expression analysis using DESeq2 (Love et al, 2014). 262 segments showing a signal ≥1 FPKM in the WT context but no significant enrichment in the *xrn1Δ* or in the *dcr1Δ* mutant were considered as putative SUTs.

For the annotation of CUTs, we used previously published RNA-Seq data from biological duplicates of *rrp6Δ* cells (Alcid & Tsukiyama, 2016). Segmentation was performed following the same procedure as described above, using a threshold of 12.96 and a sliding window of 10 nt. As no ERCC RNA spike-in was included during libraries preparation, tag densities were normalized on the total number of reads uniquely mapped on ORFs. We identified 1,280 CUTs, 116 of which overlapped >50% of transcripts defined as putative SUTs upon segmentation of our RNA-Seq data. Consequently, these 116 transcripts were not considered as SUTs.

Overall, we annotated 10 DUTs, 146 SUTs, 1,021 XUTs, and 1,280 CUTs. An lncRNA was reported as antisense when the overlap with the sense ORF was ≥1 nt.

## Small RNA-Seq

Small RNA-Seq analysis was performed from two biological replicates of YAM2478 (WT), YAM2479 (*xrn1Δ*), YAM2795 (*dcr1Δ*), and YAM2796 (*dcr1Δ xrn1Δ*) exponentially growing cells. For the control of Dcr1-GFP functionality (YAM2826 strain), only one library was prepared.

For each sample, 50 μg of total RNA were mixed with 2 μg of total RNA from the YAM2394 (WT) strain of *S. pombe* (Wery et al, 2018b), the 22–23-nt small RNAs derived from the centromeric repeats in the latter species (Djupedal et al, 2009), here constituting RNA spike-in used for the subsequent normalization of the small RNA-Seq signals.

The small RNAs (<80 nt) fraction was purified on 15% TBE–urea polyacrylamide gels. Libraries were constructed from 120 ng of purified small RNAs using the NEBNext Multiplex Small RNA Library Preparation Set for Illumina (New England Biolabs). Single-end

sequencing (50 nt) of libraries was performed on a HiSeq 2500 system (Illumina).

Adapter sequences were removed using the Atropos software (Didion et al, 2017). Reads were then mapped to the *N. castellii* and *S. pombe* reference genomes using the version 2.3.5 of Bowtie (Langmead & Salzberg, 2012), using default parameters, with no mismatch in seed alignment. Subsequent analyses used 22–23-nt reads uniquely mapped on the *N. castellii* genome. Densities were normalized on the levels of the centromeric 22–23-nt small RNAs of *S. pombe*.

### Western blot

50 μg of protein extracts were separated on a NuPAGE 4–12% Bis–Tris gel (Invitrogen) and then transferred to a nitrocellulose membrane using an iBlot Dry Blotting System (Invitrogen). The GFP and Pgk1 were detected using mouse anti-GFP (11 814 460 001, Roche) with the SuperSignal West Femto Maximum Sensitivity Substrate (Thermo Fisher Scientific) and mouse anti-Pgk1 (ab 113687; Abcam) with the SuperSignal West Pico Chemiluminescent Substrate (Thermo Fisher Scientific), respectively. Images were obtained using a ChemiDoc Imaging System (Bio-Rad).

### Microscopy

The cells were grown to mid-log phase ($OD_{600}$ 0.5) in CSM medium, at 25°C. For the live cell analysis, the cells were washed in sterile water and then loaded on a microscope slide. The images were acquired the same day with the same parameters, using a wide-field microscopy system based on an inverted microscope (TE2000; Nikon) equipped with a 100×/1.4 NA immersion objective, a CMOS camera and a collimated white light-emitting diode for the transmission. A Spectra X light engine lamp (Lumencor, Inc) was used to illuminate the samples. The whole system is piloted by the MetaMorph software (Molecular Devices). For z-stacks images, the axial (z) step is at 200 nm, and images shown are a maximum projection of z-stack images. The images were analyzed and processed using the ImageJ software.

Subcellular localization of Dcr1-GFP was performed by immunofluorescence using GFP booster/nanobody (ATTO 488; ChromoTek), according to a previously described procedure (Ries et al, 2012). Briefly, cells were loaded on concanavalin A-coated coverglass and fixed for 15 min in PBS containing 4% paraformaldehyde and 2% of sucrose. After two washes with PBS + 50 mM $NH_4Cl$, the fixed cells were blocked and permeabilized for 30 min in blocking/hybridization buffer (0.25% Triton X-100, 5% BSA, 0.004% $NaN_3$ in PBS), under gentle shaking. The cells were then labelled for 90 min with 100 μl of nanobody solution (10 μM ATTO 488 nanobody in blocking/hybridization buffer). Finally, the labelled cells were washed for 5 min in PBS, a drop of VECTASHIELD mounting medium with DAPI (Vector Labs) was added on the cells, and the coverglass was mounted on a microscope slide. Fluorescence images were acquired using the same microscope as described above. The images were analyzed and processed using the ImageJ software, as described above.

### Accession numbers and data accessibility

Sequence data generated in this work can be accessed at the NCBI Gene Expression Omnibus using accession number GSE129233. Previously published RNA-Seq data we retrieved from the Sequence Read Archive using accession number SRP056928.

Genome browsers for visualization of processed data are publicly accessible at http://vm-gb.curie.fr/castellii.

### Code accessibility

The computational scripts used in this study are accessible at https://github.com/MorillonLab/castellii.

### Availability of materials

All unique materials generated in this work are available upon request to the corresponding authors.

# Supplementary Information

# Acknowledgements

We thank David Bartel for providing the WT and *dcr1Δ* strains of *N. castellii*. We also would like to thank Patricia Legoix-Né, Virginie Raynal, Benoît Albaud, and Sylvain Baulande from the Next Generation Sequencing (NGS) platform of Institut Curie; Choumouss Kamoun from the bioinformatics platform of Institut Curie; Camille Gautier and Marc Descrimes for preliminary bioinformatics analyses and tools development; Myriam Ruault (UMR3664, Institut Curie) for assistance in microscopy analyses; Hervé Vennin-Rendos for his contribution in the construction of the pAM376 vector; Eve Samani for her enthusiastic participation in validation and characterization of yeast strains; and Ines A Drinnenberg for critical reading of the manuscript. We are grateful to all members of the lab for discussions. High-throughput sequencing was performed by the NGS platform of Institut Curie, supported by the grants ANR-10-EQPX-03 and ANR10-INBS-09-08 from the Agence Nationale de la Recherche (investissements d'avenir) and by the Canceropôle Ile-de-France. Antonin Morillon's lab is supported by the "DNA-life" grant (ANR-15-CE12-0007) from the Agence Nationale de la Recherche, and by the "EpincRNA" (starting) and "DARK" (consolidator) grants from the European Research Council.

### Author Contributions

U Szachnowski: resources, data curation, software, formal analysis, validation, investigation, and visualization.

S Andjus: formal analysis, investigation, visualization, and writing—review and editing.

D Foretek: formal analysis and investigation.

A Morillon: conceptualization, resources, supervision, funding acquisition, project administration, and writing—original draft, review, and editing.

M Wery: conceptualization, resources, data curation, formal analysis, supervision, validation, investigation, visualization, methodology,

project administration, and writing—original draft, review, and editing.

## Conflict of Interest Statement

The authors declare that they have no conflict of interest.

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
