## [Reviewer comments · Life Science Alliance]

Life Science Alliance

Endogenous RNAi pathway evolutionarily shapes the destiny of the antisense lncRNAs transcriptome

Ugo Szachnowski, Sara Andus, Dominika Foretek, Antonin Morillon, and Maxime Wery
DOI: <https://10.26508/lsa.201900407>

Corresponding author(s): Maxime Wery, Institut Curie and Antonin Morillon, CNRS-Institut Curie

Review Timeline:	Submission Date:	2019-04-26
	Editorial Decision:	2019-06-04
	Revision Received:	2019-07-22
	Editorial Decision:	2019-08-06
	Revision Received:	2019-08-21
	Accepted:	2019-08-22

Scientific Editor: Andrea Leibfried

Transaction Report:

June 4, 2019

Re: Life Science Alliance manuscript #LSA-2019-00407-T

Dr. Maxime Wery
Insitut Curie
UMR3244
26 rue d'Ulm
Paris 75248
France

Dear Dr. Wery,

Thank you for submitting your manuscript entitled "Endogenous RNAi pathway evolutionarily shapes the destiny of the antisense lncRNAs transcriptome" to Life Science Alliance. The manuscript was assessed by expert reviewers, whose comments are appended to this letter.

As you will see, the reviewers think that your results are of value to the field and they all support publication, pending revision. They provide constructive input on how to further strengthen your work and I would thus like to invite you to provide a revised version of your manuscript, addressing the individual points raised by the reviewers. Major point 1 of reviewer #1 (small RNA sequencing library normalization) is the most laborious revision point, but I trust that you can address this criticism.

The typical timeframe for revisions is three months. When submitting the revision, please include a letter addressing the reviewers' comments point by point.

Thank you for this interesting contribution to Life Science Alliance. We are looking forward to receiving your revised manuscript.

Sincerely,

Andrea Leibfried, PhD
Executive Editor

Life Science Alliance
Meyerhofstr. 1
69117 Heidelberg, Germany
t +49 6221 8891 502
e a.leibfried@life-science-alliance.org
www.life-science-alliance.org

B. MANUSCRIPT ORGANIZATION AND FORMATTING:

Reviewer #1 (Comments to the Authors (Required)):

The authors report the identification of many ncRNAs in the RNAi-competent yeast *N. castellii*, analyses of their numbers and distribution relative to *S.cerevisiae* and characterization of the effects of loss of RNA degradation factors, particularly Dcr1. The experimental and bioinformatic analyses appear to have been well performed and the major conclusions are supported by the data presented.

It was perhaps surprising, and no doubt disappointing for the authors, that the loss of Dcr1 did not have a more pronounced phenotype. However, this is a valuable contribution to the field and I am happy to recommend publication in LSA.

Minor points:

1. Figs. 1 and S1: How many DUTs? Fig. 1B indicates 8 (but also 6+4), while the text and Fig. S1D indicate 10.
2. Figure 4 and P21: The analysis is described as "immune-FISH", but does not involve hybridization or detection of nucleic acids.

Reviewer #2 (Comments to the Authors (Required)):

This study examines the role of the nuclear exosome, the exoribonuclease Xrn1p, and the endoribonuclease Dicer in regulating the antisense (as)lncRNA transcriptome in the budding yeast *Naumovozyma castellii*. It extends the findings on budding yeast lncRNA transcriptomes of Alcid and Tsukiyama (2016). Alcid and Tsukiyama examine the role of RNAi in restricting aslncRNA expression in *N. castellii* and demonstrate that a strain lacking the main catalytic subunit of the nuclear exosome (Rrp6p) grow more slowly than wild-type cells, perhaps due to the expression of aslncRNAs in the cytoplasm, which might pair with mRNAs to create a Dicer substrate. Supporting this idea, deleting the gene encoding Dicer in this nuclear exosome mutant strain partially rescues the growth phenotype. In the current study, the authors performed RNA sequencing and re-analyzed the data of Alcid and Tsukiyama. Whereas disrupting Dicer by itself had very little effect on the lncRNA transcriptome, the loss of Dicer in an *xrn1* deletion strain increased the expression of antisense Xrn1-sensitive unstable transcripts (asXUTs) and reduced growth (which contrasts to the effect of losing Dicer in the *rrp6* deletion background, previously reported by Alcid and Tsukiyama). The current study confirmed that these asXUTs are substrates of Dicer and extended the comparative analysis of aslncRNAs in different yeast species, showing that asXUTs overlap coding regions to a greater degree in *S. cerevisiae*, while antisense cryptic unstable transcripts (asCUTs) overlap coding regions to a greater degree in *N. castellii*, suggesting that the nuclear exosome-sensitive aslncRNA transcriptome has expanded in *N. castellii*. Overall, this article provides increased insight into budding yeast lncRNAs and the potential role that RNAi, the nuclear exosome, and Xrn1p have in restricting the expression of these lncRNAs in the cytoplasm.

Major concern:

1. RNA sequencing experiments were normalized using ERCC RNA spike-ins, which was appropriate, but the small RNA sequencing libraries were "normalized on tRNAs signals." Because the abundance of tRNA fragments can vary in different libraries for reasons that have nothing to do with sequencing depth, normalizing to these fragments is not appropriate. The authors should repeat the small RNA sequencing after adding spike-ins (synthetic RNAs that are ~23 nt in length) that don't map to the *N. castellii* genome. Alternatively, they could perform small RNA Northern blots for several siRNAs using the existing samples and normalize the small RNA sequencing data based on these blots.

Minor points:

1. The number of DUTs listed on the left in figure 1B disagrees with the sum of the sense and

antisense transcripts on the right and the number of DUTs listed in the main text.

2. No legends are provided to interpret the heatmaps presented in figures S2A and S2B.

3. There appears to be a discrepancy between the small RNA sequencing data presented in figures S3A and S4B, particularly in the *dcr1* deletion strains. Were the data presented in these two figures each of the two biological replicates mentioned in the methods? Do the 23 nt species from the *dcr1* deletion strain in figure S4B have other features of siRNAs?

4. The preferred first nucleotide of the 22 nt and 23 nt reads of this study (A) differs from the preferred first nucleotide of the 22 nt and 23 nt genome mapping reads in Drinnenberg et al. 2009 (U). Were reads mapping to rRNA and tRNA removed from the analyses in figures S3A and S4B? Do the small RNAs of these distributions map to the genome or are these the pre-mapped library reads?

5. In the next-to-last paragraph in the results, what do the percentages refer to ("8.1% vs 12.9%")? Also, are the asCUTs and the asXUTs plotted in figure 5E exclusively asCUTs and asXUTs (i.e., the transcripts that are not in the union of the Venn diagram in figure S1E)?

6. In the discussion, a sentence reads, "To which extent the generated small RNAs are properly loaded into Argonaute to mediate post-transcriptional gene silencing, for example at the level of translation regulation, remains unknown." Why is translational regulation proposed as the mechanism for these siRNAs that are known to be capable of directing slicing?

Reviewer #3 (Comments to the Authors (Required)):

RNA surveillance pathways play key roles in regulation of the coding and noncoding transcriptome. In particular, several classes of long noncoding RNAs are targeted for degradation by these pathways. Previous studies have identified roles for the 3'-5' exosome ribonuclease, the 5'-3' Xrn1 ribonuclease, and the RNAi pathway in processing of antisense long noncoding RNAs (aslncRNAs). This study examines the interplay these surveillance pathways in the yeast *N. castellii*, which unlike the more studied *S. cerevisiae* has a functional RNAi pathway, in addition to the exosome and Xrn1. The authors use genome-wide RNA profiling and other methods to show that (1) the exosome and Xrn1 are primarily responsible for degradation of aslncRNAs in *N. castellii*, (2) loss of the RNase III family dsRNA ribonuclease of the RNAi pathway, Dcr1, results mainly in the accumulation of Xrn1-sensitive lncRNAs, termed XUTs, (3) *dcr1* and *xrn1* mutants display synergistic growth defects, suggesting that Dcr1 becomes important in the absence of Xrn1, and (4) the exosome-sensitive antisense transcriptome in *N. castellii* is expanded relative to *S. cerevisiae*, suggesting that this yeast has adapted to the presence of cytoplasmic RNAi by increasing nuclear RNAi surveillance to prevent aslncRNA-mRNA pairs from becoming RNAi targets. The results are interesting and provide insight into adaptation strategies that allow coordination between RNAi and other RNA surveillance pathways. The comparison of two budding yeasts, one with RNAi and one without, is very powerful for this purpose. The main conclusions of the paper are supported by the results. The paper is suitable for publication in LSA.

I only have minor comments.

1. In addition to aslncRNAs (<200 nt), have the authors considered read through transcription? This

would also produce a dsRNA substrate for Dcr1 at convergently transcribed gene pairs. In this regard, overexpression of Dcr1 in *S. pombe* results in production of siRNAs from nearly all convergent transcription units (Yu et al., Mol Cell. 2014 Jan 23;53(2):262-76), suggesting that Dcr1 can target nearly all sense-antisense RNA pairs.

2. It looks like the authors detect fewer Dcr1 foci using the anti-GFP nanobody fewer fixation compared to live Dcr1-GFP imaging. Can they exclude the possibility of nuclear Dcr1 foci using their live imaging data (imaging of Dcr1-GFP with another nuclear fluorescent protein).

In summary, the results presented in this paper are valuable. The demonstration of a role for Dicer in processing of aslncRNA, and the relationship to other surveillance pathways, is important and raises the possibility that Dicer may perform this function broadly across evolution.

Please find hereafter (in italic & blue) our point-by-point answer to the comments of the three reviewers.

Reviewer #1 (Comments to the Authors (Required)):

The authors report the identification of many ncRNAs in the RNAi-competent yeast *N. castellii*, analyses of their numbers and distribution relative to *S.cerevisiae* and characterization of the effects of loss of RNA degradation factors, particularly Dcr1. The experimental and bioinformatic analyses appear to have been well performed and the major conclusions are supported by the data presented. It was perhaps surprising, and no doubt disappointing for the authors, that the loss of Dcr1 did not have a more pronounced phenotype. However, this is a valuable contribution to the field and I am happy to recommend publication in LSA.

We are grateful to reviewer #1 for the positive feedback, acknowledging the quality of our work and supporting the publication of our manuscript in LSA.

Minor points:

1. Figs. 1 and S1: How many DUTs? Fig. 1B indicates 8 (but also 6+4), while the text and Fig. S1D indicate 10.

There was an error in Figure 1B, this has been corrected. There are 10 DUTs.

2. Figure 4 and P21: The analysis is described as "immune-FISH", but does not involve hybridization or detection of nucleic acids.

We agree with the reviewer's comment. We now use the term "immunofluorescence".

Reviewer #2 (Comments to the Authors (Required)):

This study examines the role of the nuclear exosome, the exoribonuclease Xrn1p, and the endoribonuclease Dicer in regulating the antisense (as)lncRNA transcriptome in the budding yeast *Naumovozyma castellii*. It extends the findings on budding yeast lncRNA transcriptomes of Alcid and Tsukiyama (2016). Alcid and Tsukiyama examine the role of RNAi in restricting aslncRNA expression in *N. castellii* and demonstrate that a strain lacking the main catalytic subunit of the nuclear exosome (Rrp6p) grow more slowly than wild-type cells, perhaps due to the expression of aslncRNAs in the cytoplasm, which might pair with mRNAs to create a Dicer substrate. Supporting this idea, deleting the gene encoding Dicer in this nuclear exosome mutant strain partially rescues the growth phenotype. In the current study, the authors performed RNA sequencing and re-analyzed the data of Alcid and Tsukiyama. Whereas disrupting Dicer by itself had very little effect on the lncRNA transcriptome, the loss of Dicer in an *xrn1* deletion strain increased the expression of antisense Xrn1-sensitive unstable transcripts (asXUTs) and reduced growth (which contrasts to the effect of losing Dicer in the *rrp6* deletion background, previously reported by Alcid and Tsukiyama). The current study confirmed that these asXUTs are substrates of Dicer and extended the comparative analysis of aslncRNAs in different yeast species, showing that asXUTs overlap coding regions to a greater degree in *S. cerevisiae*, while antisense cryptic unstable transcripts (asCUTs) overlap coding regions to a greater degree in *N. castellii*, suggesting that the nuclear exosome-sensitive aslncRNA transcriptome has expanded in *N. castellii*. Overall, this article provides increased insight into budding yeast lncRNAs and the potential role that RNAi, the nuclear exosome, and Xrn1p have in restricting the expression of these lncRNAs in the cytoplasm.

We thank reviewer #2 for the critical reading of our manuscript and for her/his positive and constructive feedback.

Major concern:

1. RNA sequencing experiments were normalized using ERCC RNA spike-ins, which was appropriate, but the small RNA sequencing libraries were "normalized on tRNAs signals." Because the abundance of tRNA fragments can vary in different libraries for reasons that have nothing to do with sequencing depth, normalizing to these fragments is not appropriate. The authors should repeat the small RNA sequencing after adding spike-ins (synthetic RNAs that are ~23 nt in length) that don't map to the *N. castellii* genome. Alternatively, they could perform small RNA Northern blots for several siRNAs using the existing samples and normalize the small RNA sequencing data based on these blots.

*We agree that including RNA spike-in is important. As requested, we have repeated the small RNA sequencing analysis following the addition of an aliquot of total RNA from *S. pombe* in the total RNA samples from *N. castellii*. The 22-23 nt small RNAs derived from the centromeric repeats of *S. pombe* constitute the RNA spike-in used as the reference for the normalization of the small RNA-Seq signals. The results of the new experiment upon normalization on the spike-in signals are very similar to the previous data, based on the normalization on the tRNAs signals. In fact, upon spike-in normalization, we observed that the tag densities for the tRNAs are globally unaffected in the different samples (see Figure R1 above).*

Figure R1. Densities for *N. castellii* tRNAs upon normalization of the small RNA-Seq signals on the *S. pombe* small RNA spike-in.

Scatter plots showing the densities (tag/nt, \log_2 scale) of 22-23 nt reads for protein-coding genes (grey dots) and tRNAs (red dots) in the WT, *xrn1*Δ, *dcr1*Δ and *xrn1*Δ *dcr1*Δ conditions, upon normalization on the RNA spike-in signals (22-23 nt reads mapped to *S. pombe* centromeric repeats). The red line indicates no change (condition1/condition2 ratio = 1).

Thus, the new small RNA sequencing analysis confirms our initial conclusions, which remain unchanged. Again, we are grateful to reviewer #2 for her/his comment which allowed us to reinforce our conclusions, in a more robust and rigorous manner.

Minor points:

1. The number of DUTs listed on the left in figure 1B disagrees with the sum of the sense and antisense transcripts on the right and the number of DUTs listed in the main text.

There was an error in Figure 1B and this has been corrected. There are 10 DUTs.

2. No legends are provided to interpret the heatmaps presented in figures S2A and S2B.

A scale has been added.

3. There appears to be a discrepancy between the small RNA sequencing data presented in figures S3A and S4B, particularly in the *dcr1* deletion strains. Were the data presented in these two figures each of the two biological replicates mentioned in the methods? Do the 23 nt species from the *dcr1* deletion strain in figure S4B have other features of siRNAs?

*The results shown in the previous Fig S3A and S4B corresponded to distinct datasets (ie different libraries & sequencing). This might explain (at least partly) the minor variations in terms of first base distribution between the two datasets. These figures have been updated in the revised version of the manuscript, as the small RNA-Seq analysis has been repeated (response to major concern). The new libraries for the WT, *xrn1*Δ, *dcr1*Δ, *xrn1*Δ *dcr1*Δ and Dcr1-GFP conditions were constructed and sequenced in parallel, so that the profile obtained for the Dcr1-GFP strain (new Fig S4C) can be directly compared to the profile of the WT and *dcr1*Δ strains (Fig S3A). The small 23 nt peak detected in *dcr1*Δ in the previous Fig S4B, but not in the previous Fig S3A, is absent in the new dataset.*

4. The preferred first nucleotide of the 22 nt and 23 nt reads of this study (A) differs from the preferred first nucleotide of the 22 nt and 23 nt genome mapping reads in Drinnenberg et al. 2009 (U). Were reads mapping to rRNA and tRNA removed from the analyses in figures S3A and S4B? Do the small RNAs of these distributions map to the genome or are these the pre-mapped library reads?

*In the initial version of the manuscript, Fig S3A and S4B used reads that uniquely mapped to the *N. castellii* genome, with no additional filter to remove the reads matching rRNAs (unlikely as they are repeated sequences) and tRNAs (used for normalization of the signals).*

In the revised manuscript, the small RNA-Seq analysis has been repeated (response to major comment), and the reads matching to rRNAs and tRNAs were filtered out. In the new Fig S3A and S4C (previously S4B), the preferred first base of the 22-23 nt small RNAs now appears to be 'U', which is consistent with what has been previously reported by Drinnenberg et al (2009).

5. In the next-to-last paragraph in the results, what do the percentages refer to ("8.1% vs 12.9%")? Also, are the asCUTs and the asXUTs plotted in figure 5E exclusively asCUTs and asXUTs (i.e., the transcripts that are not in the union of the Venn diagram in figure S1E)?

*These percentages correspond to the global coverage of the coding transcriptome by aslncRNAs in *N. castellii* and *S. cerevisiae*. This has been clarified in the main text.*

*The numbers in Fig 5E correspond to the full set of asCUTs or asXUTs in each species (ie 868 asCUTs and 622 asXUTs in *N. castellii*), including those that overlap an asXUT or an asCUT, respectively.*

6. In the discussion, a sentence reads, "To which extent the generated small RNAs are properly loaded into Argonaute to mediate post-transcriptional gene silencing, for example at the level of translation regulation, remains unknown." Why is translational regulation proposed as the mechanism for these siRNAs that are known to be capable of directing slicing?

This was fully speculative. In absence of experimental data supporting this hypothesis, the sentence has been modified to avoid confusion, now only stating: "To which extent the generated small RNAs are properly loaded into Argonaute to mediate post-transcriptional gene silencing remains unknown."

Reviewer #3 (Comments to the Authors (Required)):

RNA surveillance pathways play key roles in regulation of the coding and noncoding transcriptome. In particular, several classes of long noncoding RNAs are targeted for degradation by these pathways. Previous studies have identified roles for the 3'-5' exosome ribonuclease, the 5'-3' Xrn1 ribonuclease, and the RNAi pathway in processing of antisense long noncoding RNAs (aslncRNAs). This study examines the interplay these surveillance pathways in the yeast *N. castellii*, which unlike the more studied *S. cerevisiae* has a functional RNAi pathway, in addition to the exosome and Xrn1. The authors use genome-wide RNA profiling and other methods to show that (1) the exosome and Xrn1 are primarily responsible for degradation of aslncRNAs in *N. castellii*, (2) loss of the RNase III family dsRNA ribonuclease of the RNAi pathway, Dcr1, results mainly in the accumulation of Xrn1-sensitive lncRNAs, termed XUTs, (3) dcr1 and xrn1 mutants display synergistic growth defects, suggesting that Dcr1 becomes important in the absence of Xrn1, and (4) the exosome-sensitive antisense transcriptome in *N. castellii* is expanded relative to *S. cerevisiae*, suggesting that this yeast has adapted to the presence of cytoplasmic RNAi by increasing nuclear RNAi surveillance to prevent aslncRNA-mRNA pairs from becoming RNAi targets. The results are interesting and provide insight into adaptation strategies that allow coordination between RNAi and other RNA surveillance pathways. The comparison of two budding yeasts, one with RNAi and one without, is very powerful for this purpose. The main conclusions of the paper are supported by the results. The paper is suitable for publication in LSA.

We thank reviewer #3 for his/her positive feedback and for supporting the publication of our manuscript in LSA.

I only have minor comments.

1. In addition to aslncRNAs (<200 nt), have the authors considered read through transcription? This would also produce a dsRNA substrate for Dcr1 at convergently transcribed gene pairs. In this regard, overexpression of Dcr1 in *S. pombe* results in production of siRNAs from nearly all convergent transcription units (Yu et al., Mol Cell. 2014 Jan 23;53(2):262-76), suggesting that Dcr1 can target nearly all sense-antisense RNA pairs.

*This is a very interesting comment. We did not systematically analyze the possibility that convergent mRNAs could form dsRNA that would be used by Dcr1 as a substrate for small RNA production. If this appears to be frequent in fission yeast (upon Dcr1 overexpression), navigating across the *N. castellii* genome using the browser provided with this article indicates that it is not the case in the budding yeast species. As representative examples, see the two snapshots provided in the manuscript, showing the absence of 22-23 nt small RNAs produced from the convergent C05770/C05780 mRNAs (Fig 3B) and A12450/A12460 mRNAs (Fig S3E).*

2. It looks like the authors detect fewer Dcr1 foci using the anti-GFP nanobody fewer fixation compared to live Dcr1-GFP imaging. Can they exclude the possibility of nuclear Dcr1 foci using their live imaging data (imaging of Dcr1-GFP with another nuclear fluorescent protein).

*We did not systematically compare the number of Dcr1 foci per cell detected in fixed cells using the anti-GFP nanobody and by direct GFP visualization in living cells. We conclude that Dcr1 localizes in the cytoplasm in *N. castellii*, but as mentioned in the discussion, we cannot exclude "the possibility*

that a small amount of Dcr1 molecules in the cell localize in the nucleus, into levels that are under the detection threshold of our microscope”.

In summary, the results presented in this paper are valuable. The demonstration of a role for Dicer in processing of aslncRNA, and the relationship to other surveillance pathways, is important and raises the possibility that Dicer may perform this function broadly across evolution.

Again, we are grateful to reviewer #3 for supporting our work.

August 6, 2019

RE: Life Science Alliance Manuscript #LSA-2019-00407-TR

Dr. Maxime Wery
Insitut Curie
UMR3244
26 rue d'Ulm
Paris 75248
France

Dear Dr. Wery,

Thank you for submitting your revised manuscript entitled "Endogenous RNAi pathway evolutionarily shapes the destiny of the antisense lncRNAs transcriptome". As you will see, reviewer #2 re-assessed your manuscript and appreciates the introduced changes. We would be happy to publish your paper in Life Science Alliance pending final revisions necessary to meet our formatting guidelines:

- please provide the code used or deposit it to github
- the genome browser link for visualization purposes seems very useful, it would be good to provide the underlying code for conservation purposes as well.

A. FINAL FILES:

-- Summary blurb (enter in submission system): A short text summarizing in a single sentence the study (max. 200 characters including spaces). This text is used in conjunction with the titles of

papers, hence should be informative and complementary to the title. It should describe the context and significance of the findings for a general readership; it should be written in the present tense and refer to the work in the third person. Author names should not be mentioned.

B. MANUSCRIPT ORGANIZATION AND FORMATTING:

Sincerely,

Reviewer #2 (Comments to the Authors (Required)):

The authors have satisfactorily addressed my concerns.

August 22, 2019

RE: Life Science Alliance Manuscript #LSA-2019-00407-TRR

Dr. Maxime Wery
Insitut Curie
UMR3244
26 rue d'Ulm
Paris 75248
France

Dear Dr. Wery,

Thank you for submitting your Research Article entitled "Endogenous RNAi pathway evolutionarily shapes the destiny of the antisense lncRNAs transcriptome". It is a pleasure to let you know that your manuscript is now accepted for publication in Life Science Alliance. Congratulations on this interesting work.

DISTRIBUTION OF MATERIALS:

Again, congratulations on a very nice paper. I hope you found the review process to be constructive and are pleased with how the manuscript was handled editorially. We look forward to future exciting submissions from your lab.

Sincerely,

Andrea Leibfried, PhD
Executive Editor
Life Science Alliance
Meyerohofstr. 1
69117 Heidelberg, Germany
t +49 6221 8891 502
e a.leibfried@life-science-alliance.org
www.life-science-alliance.org